# Hierarchical modelling of immunoglobulin coated bacteria in dogs with chronic enteropathy shows reduction in coating with disease remission but marked inter-individual and treatment-response variability

Lina María Martínez-López[1], Alexis Perez-Gonzalez[2], Elizabeth Ann Washington[3], Andrew P. Woodward[4], Alexandra Jazmin Roth-Schulze[5], Julien R. S. Dandrieux[1], Thurid Johnstone[1], Nathalee Prakash[1¤], Aaron Jex[3,4], Caroline Mansfield[1]*

1 Department of Veterinary Clinical Sciences, Melbourne Veterinary School, The University of Melbourne, Werribee, Victoria, Australia, 2 Melbourne Cytometry Platform, Melbourne Dental School, The University of Melbourne, Carlton, Victoria, Australia, 3 Veterinary Biosciences, The University of Melbourne, Parkville, Victoria, Australia, 4 Faculty of Veterinary and Agricultural Sciences, The University of Melbourne, Werribee, Victoria, Australia, 5 Walter and Eliza Hall Institute of Medical Research, Parkville, Victoria, Australia

¤ Current address: Mount Pleasant Veterinary Centre, Gelenggang, Singapore
* cmans@unimelb.edu.au

## Abstract

Chronic enteropathies are a common problem in dogs, but many aspects of the pathogenesis remain unknown, making the therapeutic approach challenging in some cases. Environmental factors are intimately related to the development and perpetuation of gastrointestinal disease and the gut microbiome has been identified as a contributing factor. Previous studies have identified dysbiosis and reduced bacterial diversity in the gastrointestinal microbiota of dogs with chronic enteropathies. In this case-controlled study, we use flow cytometry and 16S rRNA sequencing to characterise bacteria highly coated with IgA or IgG in faecal samples from dogs with chronic enteropathy and evaluated their correlation with disease and resolution of the clinical signs. IgA and IgG-coated faecal bacterial counts were significantly higher during active disease compared to healthy dogs and decreased with the resolution of the clinical signs. Characterisation of taxa-specific coating of the intestinal microbiota with IgA and IgG showed marked variation between dogs and disease states, and different patterns of immunoglobulin enrichment were observed in dogs with chronic enteropathy, particularly for *Erysipelotrichaceae*, *Clostridicaceae*, *Enterobacteriaceae*, *Prevotellaceae* and *Bacteroidaceae*, families. Although, members of these bacterial groups have been associated with strong immunogenic properties and could potentially constitute important biomarkers of disease, their significance and role need to be further investigated.

**Data Availability Statement:** The data underlying the results presented in the study are available from GitHub: https://github.com/Lina-Maria/Immunoglobulin_coating_CE_dogs.

**Funding:** This work was supported by the International and Postgraduate Research Scholarship (IPRS) from The Australian Government 2014 (LM). This work was funded by The Comparative and Gastroenterology Society/Waltham Grant 2016 (LM) [https://vetmed.tamu.edu/cgs/about/] and Early Career Researcher funding through the University of Melbourne 2018 (CM). The sponsors played no role in study design, data collection/analysis and were not involved in preparation of the publication or decision to publish.

**Competing interests:** The authors have declared that no competing interests exist.

# Introduction

Chronic enteropathy (CE) in dogs constitutes a group of disorders that results in gastrointestinal (GI) tract inflammation and persistent or recurrent gastrointestinal clinical signs; CE is often termed inflammatory bowel disease (IBD) when immunosuppressive therapy is required [1, 2]. Although, CE is considered to be a frequent cause of veterinary consultation, due to a lack of clinical specificity classification is made retrospectively based more on the response to treatment rather than based on the pathogenesis of the disease; leading to the administration of treatments that, in some cases, are lengthy or produce a short-term remission and relapse of the clinical signs [2, 3].

In CE, it is considered that the triad of host genetics- immune system- microenvironment, particularly dietary antigens, and the gastrointestinal flora, are closely related to the development of gastrointestinal inflammation [4]. For example, it is proposed that either inappropriate responses to normal gut microbiota lead to excessive immunological stimulation or; compositional changes in the gut microbiota elicit pathological immune responses [5]. Immunoglobulins (Igs) are part of the adaptive immune system and constitute crucial pathways that directly influence the function and structure of the microbiota [6]. Although intestinal IgA is the predominant Ig isotype produced in the intestine, other Igs such as IgM and IgG can also be produced, and relative levels of bacteria coated with Igs may correlate with the magnitude of the inflammatory response triggered by specific intestinal bacterial species [7].

Few studies related to Igs have been done in dogs with CE, with varied results. One study identified increased numbers of plasma cells in the intestinal mucosa (particularly IgA+ and IgG+ cells) [8]. An earlier study identified higher mucosal concentrations of IgA and IgG in dogs with CE compared to healthy dogs [9], whereas another observed no difference in plasma cell populations in the mucosa of dogs with CE compared to healthy dogs (although they detected a lower concentration of IgA in the blood of dogs with CE) [10]. Maeda et al. reported decreased IgA concentrations in the faeces, duodenum, and peripheral mononuclear cells of dogs with IBD [11]. The discordance in results could be due to differing methodologies, sample sites (e.g., duodenum *versus* colon), different disease aetiologies (e.g., small intestinal bacterial overgrowth (SIBO), antibiotic-responsive diarrhoea (ARD), or IBD) and the breed of dogs studied. More recently, Soontararak *et al*. concluded that dogs with IBD had significantly higher percentages and overall amounts of IgG and IgA bound to their intestinal bacteria than healthy dogs [12]. However, time-series analysis and studies in diet-responsive dogs (DRE) and ARE dogs have not been performed to date.

Here, we investigated the IgA and IgG coating of faecal bacteria in healthy dogs and dogs with CE and the correlation with disease stage and resolution of the clinical signs. We hypothesised that (1) dogs with CE have a higher proportion of highly IgA-coated bacteria in their faeces compared to healthy dogs; (2) flow cytometry and 16S rRNA could identify members of the microbiota that impact disease susceptibility or severity in dogs with CE and (3) resolution of the clinical signs in dogs with CE is associated with the eradication of bacteria highly coated with IgA.

# Materials and methods

## Study dogs

Dogs with signs of chronic GI disease (> 3 weeks), including persistent and/or recurrent vomiting and/or diarrhoea and/or weight loss, presented to the U-Vet Werribee Animal Hospital at the University of Melbourne were enrolled into the prospective study (S1 Table).

Dogs underwent complete clinical evaluation, with the exclusion of extra-intestinal disease, prior to enrolment in the study by a combination of faecal analysis (faecal flotation, faecal polymerase chain reaction [PCR] and faecal cytology), blood testing (including haematology, comprehensive serum biochemistry, canine pancreatic lipase immunoreactivity, cobalamin, serum cortisol +/- adrenocorticotropic hormone (ACTH) stimulation test and canine trypsin-like immunoreactivity) and abdominal ultrasound. Dogs were not included in the trial if there was a history of dietary or medical therapy three weeks prior to analysis or if hypoalbuminemia (albumin < 20 g/L) was present.

The gastrointestinal disease activity was scored using the canine chronic enteropathy clinical activity index (CCECAI) [13].

The treatment for the study consisted first of a diet trial (hydrolysed or hypoallergenic [limited number of protein sources] prescription diets determined on previous dietary history) for at least two weeks. If the dog showed improvement of the clinical signs after this period (>75% reduction in CCECAI), dietary treatment continued, and the dog was classified as having diet-responsive enteropathy (DRE) if remission was maintained for at least 6 weeks. If no response to treatment was observed, antibiotic treatment, consisting of 10mg/kg of oxytetracycline twice a day for two weeks, was added to the dietary therapy. Dogs that responded were classified as having antibiotic-responsive enteropathy (ARE). If they did not respond, dogs were additionally prescribed prednisolone at a dose of 2mg/kg once per day for ten days, followed by a tapering protocol, with these dogs being classified as having immunosuppressant-responsive enteropathy (IRE) [14]. Clinical remission was defined as a minimum of six weeks without clinical signs.

**Controls.** Healthy client-owned dogs with no signs of gastrointestinal disease, no antibiotic or other drug treatment (including non-steroidal anti-inflammatory or proton pump inhibitors) or change of diet in the previous three months were recruited. Detailed information regarding diet (type, treats, and changes within the previous three months, coprophagia, pica), health status, previous diseases, travel history, level and type of exercise, body condition score, and increase or decrease in body weight in the previous three months was collected. All dogs were fed with commercial diets (S2 Table).

All animal procedures were done in accordance with the Animal Ethics committee of the University of Melbourne (Animal Ethics Committee approval AEC # 1112072.2 and AEC #1413272.1). Owners gave written consent and were able to withdraw their animals from the trial at any point.

## Samples

**Faeces.** Serial stool samples (at the beginning, during treatment and remission periods at different time points) were collected upon voiding without contacting the environment (to avoid transfer of genetic material) or via rectal examination and placed immediately in a sterile container at 4°C. Aliquots of 250 mg were stored within four hours after collection at -80°C, until further analysis.

*Control dogs.* Two samples were collected, one month apart and aliquoted in the same way as described above.

**Histology.** Endoscopic biopsies of the stomach, duodenum, and colon before and after treatment were placed in 10% neutral-buffered formalin, routinely processed, and stained with haematoxylin and eosin (H&E). Specimens were evaluated by a veterinary pathologist with experience in gastrointestinal pathology and scored using published international guidelines (World Small Animal Veterinary Association [WSAVA]) [15]. Location and severity of changes were noted for each dog.

## Flow cytometry and sorting of IgA$^+$ and IgA$^-$ bacteria

All buffers were prepared freshly under sterile conditions. One aliquot of faeces was placed in a sterile conical tube and 2.5 mL of cold (4˚C), sterile, filtered (0.22 μm, Millipore®) phosphate-buffered saline (PBS) 1X was added (phosphate-buffered saline 10X concentrate, P5493 Sigma®). Samples were incubated on ice for 1 hour and homogenised by vortexing every 15 minutes for 1 minute until the faecal material was completely dissolved. Next, samples were centrifuged at 40 x $g$ for twenty minutes at 4˚C to separate larger faecal particles from bacteria. Supernatants were passed through a 70 μm sterile filter (Cell strainer, Z742103 Sigma®) into a new, sterile tube. Aliquots of 500 μL were stored at -80˚C. 10 μL of the faecal bacterial supernatant was diluted in 500 μL of 5 μM tris buffer containing 5 μM SYTO 17 (SYTO™ 17 Red Fluorescent Nucleic Acid Stain—5 μM solution in DMSO, catalogue number: S7579, Invitrogen™) and bacteria were counted, using 50 μL of the CountBright™ Absolute Counting Beads, by flow cytometry (Catalogue number: C36950) on a FACSVerse (BD Biosciences). The instrument was calibrated daily with BD FACSuite CS&T Research Beads (BD Biosciences). The cell-permeant SYTO 17 was used to (1): set appropriate side-scattered (SSC) gains and threshold levels during the experimental setup (appropriate to define combinations of gain and threshold allowing the full visualisation of all bacteria cells while limiting the noise) and (2) for gating purposes limiting the analysis of Ig $^+$ or−events to nucleic acid binding cells.

Antibodies for detecting IgA and IgG on bacteria and their subsequent sorting were first titrated and tested using the FACSVerse. Goat anti-dog IgA-fluorescein isothiocyanate (FITC) (Serotec SEAA131F, Abacus ALS) or sheep anti-dog IgG-FITC (Serotec SEAA132F, Abacus ALS) were titrated at 1:50, 1:100, 1:200 and 1:400. SYTO 17 was used to identify bacterial cells and tested at the following concentrations: 1 μM, 5 μM and 20 μM. The highest concentration resulted in bacterial death. A total of 10,000 events, collected at a flow rate of 1,000 events per second, were analysed. Fluorescence signals were evaluated bi-exponentially, and SSC logarithmically. Antibody concentrations of 1:200 and a Syto17 concentration of 5 μM were determined to be optimal.

Samples for sorting were prepared from the aliquots stored at -80˚C; $10^7$ bacteria were washed with a 1 mL sterile and filtered staining buffer consisting of PBS 1X supplemented with 1% (w/v) bovine serum albumin (BSA, A7030 Sigma®) and centrifuged for 5 minutes (8000 x $g$, 4˚C). A sample of this bacterial suspension was saved (100 μL) as the input (total bacteria-pre-sort) sample for 16S rRNA gene sequencing analysis (before centrifugation). After centrifugation, the supernatants were removed, and the pellets were resuspended in 1 mL of staining buffer and centrifuged again for 5 minutes (8000 x $g$, 4˚C). Supernatants were then removed, and the bacterial pellets were resuspended in 100 μL of blocking buffer (staining buffer containing 20% normal goat serum for IgA (G9023 Sigma®) or staining buffer containing 20% normal sheep serum for IgG (S3772 Sigma®), incubated for 20 minutes on ice, and then stained with 100 μL of staining buffer containing goat anti-dog IgA-FITC 1:200 or sheep anti-dog IgG-FITC 1:200. Samples were incubated for 30 minutes at 4˚C in the dark. Samples were then washed twice with 1 mL of staining buffer and centrifuged for 10 minutes (8000 x $g$, 4˚C). Finally, the bacterial pellet was resuspended in 500 μL of 5 μM sterile tris buffer containing 5 μM SYTO 17 and kept at 4˚C in the dark for sorting.

Cell sorting was carried out on a MoFlo™ Astrios EQ (Beckman Coulter, Inc). The sorter was calibrated with Astrios QC beads (Beckman Coulter, Inc) for the optical alignment and Flow-Check™ Pro Fluorospheres (Beckman Coulter, Inc) to confirm drop charge delay values. Laser and emission filters were 488 nm and 526/52 for FITC and 640 nm and 671/30 for SYTO 17, respectively. The MoFlo Astrios EQ small particle detection module was slightly modified to maximise bacterial forward scatter detection via the removal of both the 50/50 scatter beam

splitter cube and the neutral density filter placed before the FSC photomultipliers (PMT). The optical alignment of the MoFlo Astrios EQ FSC small particle detector module was optimised for maximum signal output while running LX200 nm beads (Beckman Coulter). Bacterial sample acquisition was triggered on 488 nm SSC with optimum PMT voltage and threshold settings set to minimise SSC noise while allowing the detection of all SYTO 17$^+$ bacteria. Sort gates were based on bacterial forward and side scattering properties, followed by their ability to bind the DNA dye SYTO 17. Immunoglobulin positive (Ig$^+$) and negative (Ig$^-$) populations were then identified by their ability to bind fluorescently labelled antibodies. For cell sorting, stringent gates were applied to avoid cross contaminating the lower frequency Ig$^+$ sort products with Ig$^-$ cells. A total of 100,000 events from each population were sorted. Each fraction was stored at -80˚C prior to PCR and sequencing of bacterial 16S rRNA gene.

Multiple precautions were taken to minimise potential contamination of sorted samples. Sterile filtered PBS 1X was used for sheath fluid, and the flow cytometer was sterilised according to the manufacturer's recommended protocol, with the sheath fluid filter replaced routinely. Before commencing the cell sorting and between samples, the sample line was washed with sodium hypochlorite 2.5% for 2 minutes, followed by sterile filtered PBS 1X for 4 minutes. In addition, samples of sheath fluid were collected directly from the jet and immediately before each sorting to assess any potential contaminants in the fluid lines.

Every run was done using samples from both healthy dogs and dogs with CE. All visits from each dog were sorted the same day. FlowJo v10.0.8 (Becton Dickinson) was used during the post-acquisition analysis of IgG$^+$ and IgA$^+$ SYTO 17$^+$ bacteria using data files generated in the MoFlo Astrios EQ during sorts.

## Bacterial 16S rRNA gene analysis

The V4 hypervariable region of the bacterial 16S rRNA gene was PCR-amplified with primers 515F-OH1 (`GTGACCTATGAACTCAGGAGTCGGACTACNVGGGTWTCTAAT`) and 806R-OH2 (`CTGAGACTTGCACATCGCAGCGTGYCAGCMGCCGCGGTAA`); 2.5 μL were added directly to a PCR master mix (20 μL reaction/sample). This primer pair amplifies the region 533–786 in the *Escherichia coli* strain 83972 sequence (Greengenes accession no. prokMSA_id:470367). Cycling conditions consisted of 95˚C for 3 minutes followed by 35 cycles of 95˚C for 15 seconds, 60˚C for 30 seconds, 72˚C for 30 seconds and 72˚C for 7 minutes. A 10 min 95˚C step at the beginning of the PCR was added to heat lyse the bacteria. Individual "barcode" sequences of 8 base pairs were added to each sample so they could be distinguished and sorted during data analysis. Specificity and amplicon size were verified by gel electrophoresis, and the amplicons were checked and measured using the Agilent High Sensitivity DNA assay in Agilent 2100 Expert (samples for checking were chosen randomly). The 600-cycle kit was used for paired-end sequencing (2x 311 cycles) using Illumina MiSeq. Due to the number of samples (700 samples run in duplicates), samples were sequenced in two runs (Run 1: 728 samples, Run 2: 672 samples). The sequencing depth was ~14,000 reads per sample.

Raw data were analysed using the open-source software package Quantitative Insights into Microbial Ecology (QIIME). Version 1 (QIIME1, release 1.9.0) was used to extract the barcodes using the script extract_barcodes.py [16]. Subsequent steps used version 2 (QIIME2, release 2018.11) [17]. Demultiplexing was carried out using the Qiime2 plugin: qiime demux emp-paired and quality filtering using the pipeline DADA2 [18]. The plugin qiime dada2 denoise-paired merges, denoises and clusters paired-end reads based on amplicon sequences variants ([ASV] threshold 100% similarity). Taxonomy assignment to the unique sequences was done by using a pre-trained naïve Bayes classifier trained against Greengenes (13_8 revision) trimmed to contain only the V4 hypervariable region and pre-clustered at 99% sequence

identity. A phylogenetic tree was generated using sep from the q2-fragment-insertion plugin [19]. Samples with less than 500 reads (counts) and features (taxa) with a total abundance (summed across all samples) of less than ten were removed. For posterior analysis, runs and duplicates were merged.

The open-source R package called Decontam (version 1.11.1) was used to identify and remove external contaminants in the sequencing data [20] employing the "Prevalence" contaminant identification method. In this method, the prevalence (presence/absence across samples) of each sequence feature in true positive samples is compared to the prevalence in negative controls to identify contaminants. Negative controls included samples of water collected from the sheath fluid before each sorting, nuclease-free water and tris-buffer used in the resuspension of the bacterial pellet for the faecal samples. Based on the distribution scores, a threshold of 0.5 was chosen. The sequencing run was included as a factor in the model. After applying this method, 221 ASVs were removed, leaving 10609. Then, we pruned ASVs that were not present or were present as singletons, with 7335 taxa remaining.

Measurements of alpha- and beta-diversity were done using Microbiome (version 1.10.0), Vegan (version 2.5.6) and Phyloseq (version 1.32.0) packages from R [21]. Alpha-diversity was calculated using the Shannon index. Faecal cell sortings were rarefied at 1000 sequences per sample. Beta-diversity was assessed using the Aitchison distance [22, 23]. Here, a pseudocount of 1 was assigned to facilitate the transformation (since the log of zero is undefined). Then, the centred log-ratio (CLR) of the transformed counts was calculated.

Hierarchical clustering was performed using UPGMA (unweighted pair group method with arithmetic mean) method (hclust parameter method = "average") and the Bray-Curtis distance.

## Statistical analyses

**Flow cytometry.** The flow cytometry response data comprised counts of Ig-bound and unbound cells. The counts represented the number of positive cells counted out of a total number of counted cells, so they can be considered independent draws from a binomial distribution with $n$ trials, where the binomial probability on each trial $p$ is unknown and is to be estimated from the data. The hierarchical model for the proportion of positive cells was therefore defined, in Wilkinson notation, as:

$$\mathrm{logit}(y) \sim 1 + Ig*disease*treatment + (1 + Antibody|Dog).$$

The '*' operator indicates interacting terms, which in this case included all combinations of the categorical variables Ig, disease (classification) and treatment (stage). The hierarchical intercept term (1|Dog) accommodates inter-subject variation. The model was implemented in the 'brms' package in R [24]. The package's default flat priors for the regression coefficients were retained, and the default priors also retained for the model intercept as well as the default hyperpriors for the variance parameters [24]. Convergence of the model was assessed using the Gelman-Rubin diagnostic [25] and effective sample sizes, and the precision of parameter estimation expressed as 90% credible intervals. Informativeness of the final model was expressed using the marginal and conditional coefficients of determination [26] as implemented in package 'performance' (version 0.7.1). The posterior predicted relative abundances under various conditions were visualised to display the model results.

**Diversity.** Shannon index (alpha diversity) was defined as the response in a linear mixed model, which included a subject-level random intercept using the following formula: *log(Shannon) ~ 1 + Ig * disease * treatment + (1 | Dog)*. The '*' operator indicates interacting terms, which in this case included all combinations of the categorical variables Ig, disease, and

treatment. The hierarchical intercept term (1|Dog) accommodates inter-subject variation. The model was defined using the 'lme4' package (version 1.1.23) in R [27]. Informativeness of the model was assessed using the marginal and conditional coefficients of determination as implemented in the 'muMin' package (version 1.43.17) from R [26, 28]. The package 'emmeans' (version 1.5.0) from R was used for posthoc comparisons among groups and stages and for estimating marginal means and their 95% confidence intervals [29].

Differences in beta diversity were calculated in rarefied data based on Permutational Multivariate Analysis of Variance using Distance Matrices (the Aitchison distance) and the function Adonis from the program Vegan 2.5.6. in R [30].

**Taxon abundance.** For determination of the taxon abundance, a hierarchical model was implemented to address the following: (1) the effect of immunoglobulin-positive or immunoglobulin-negative binding status on the relative abundance of bacterial taxa; (2) the effect of disease type on relative abundance, and differential effects of immunoglobulin binding conditional on disease type; (3) the change in relative abundance after treatment, and differential effects of treatment given disease type and Ig binding status; and (4) between-subject variability in relative bacterial abundance, under all conditions.

The proposed model for implementing these requirements included: (1) a multinomial response distribution, with one response category for each bacterial taxon, and the sample-level total number of trials; (2) a subject-level random intercept, expressing the between-subject variability in relative abundance; (3) categorical predictors and interaction terms for disease type, immunoglobulin fraction, and treatment stage.

The model for the proportional abundance of the *mth* taxon, for either the IgA or IgG dataset, was defined, in Wilkinson notation, as:

$$logit(y\_m) \sim 1 + Ig * disease * treatment + (1|Dog).$$

The '*' operator indicates interacting terms, which in this case included all combinations of the categorical variables Ig, disease, and treatment. The full model for 11 selected taxa, and an 'other' category comprising the sum of the remaining counts, was implemented using the multinomial response distribution in the 'brms' package in R [24]. The package's default prior for the intercepts and hyperpriors of the variance terms were retained, and priors for the regression coefficients were all specified as normal with mean zero and standard deviation 5, N(0,5), intended to be minimally informative. The convergence and precision of the model were assessed as for the flow cytometry model above. The posterior predicted relative abundances under various conditions were visualised to display the model results.

The effect of immunoglobulin binding was determined using the ratio of the predicted proportional abundance in the immunoglobulin-bound condition to the baseline condition, such that values greater than 1 indicated enrichment in the immunoglobulin-bound sample (highly coated Ig) and values lower than 1 indicated decline in the immunoglobulin-bound sample. This quantity is related to the 'Palm index' [7], but obtained from posterior prediction rather than the raw data. Posterior distributions of this ratio were obtained from the Markov chain Monte Carlo (MCMC) samples.

## Results

### Study dogs

To assess the stability and the dynamics of the Ig coating in healthy dogs, faecal samples from eleven adult, client-owned, dogs with no signs of gastrointestinal disease were collected (Samples collected between 2015–2017). Two samples were collected from each dog, one month

apart (Diagram 1). The group consisted of four males (one entire) and seven females (one entire) of eight different breeds and aged between one and eleven years old (S2 Table).

Study samples were collected in two phases: phase one between 2012 and 2014 and phase two between 2015–2017, correlating to two clinical trials. One animal withdrew from the study, and only provided samples during active disease (CE 3) (S1 Table). Two samples per dog were collected during phase one, the first one at the time of diagnosis (active disease) and another after resolution of the clinical signs (remission period). During phase two, faecal samples from dogs with CE were also collected during active disease and over time whilst being treated until the remission of clinical signs. Twelve dogs were recruited during the period 2012–2014, and ten dogs during the later study (S1 Table). The affected dogs consisted of 12 males (two entire) and ten females (one entire), with 15 different breeds and age between one and ten years old. Of the 22 dogs analysed, eleven had DRE, eight had ARE and three required immunosuppression (IRE) (Fig 1). The impact of treatment on disease was monitored using the CCECAI [13]. Although it has been suggested that younger dogs with less severe disease, and a predominance of large intestinal signs, are more likely to respond to elimination diet alone [13, 31], we did not observe a clear pattern, possibly due to the number of dogs included in the study (Table 1).

**Immunoglobulin coating of faecal bacteria in healthy (control) dogs.** The proportion of bacteria coated with IgA and IgG was determined during bacterial cell sorting analysis. For the first visit ('before'), the average estimated proportion of Ig-coated bacteria was 28% for IgA (IgA$^+$) and IgG (IgG$^+$). For the second visit ('after'), the average estimated proportion was 25% for IgA$^+$ and 24% for IgG$^+$ (Fig 2 and S3 Table).

Data are expressed as a proportion of bacterial cells labelled with anti-dog-FITC-IgA or IgG relative to the total cell population stained with SYTO 17. DRE: Diet-responsive enteropathy. ARE: Antibiotic-responsive enteropathy. IRE: Immunosuppressant-responsive enteropathy. 'Before' corresponds to V1 in healthy dogs and active disease in CE dogs. 'After' corresponds to V2 in healthy dogs and remission in CE dogs.

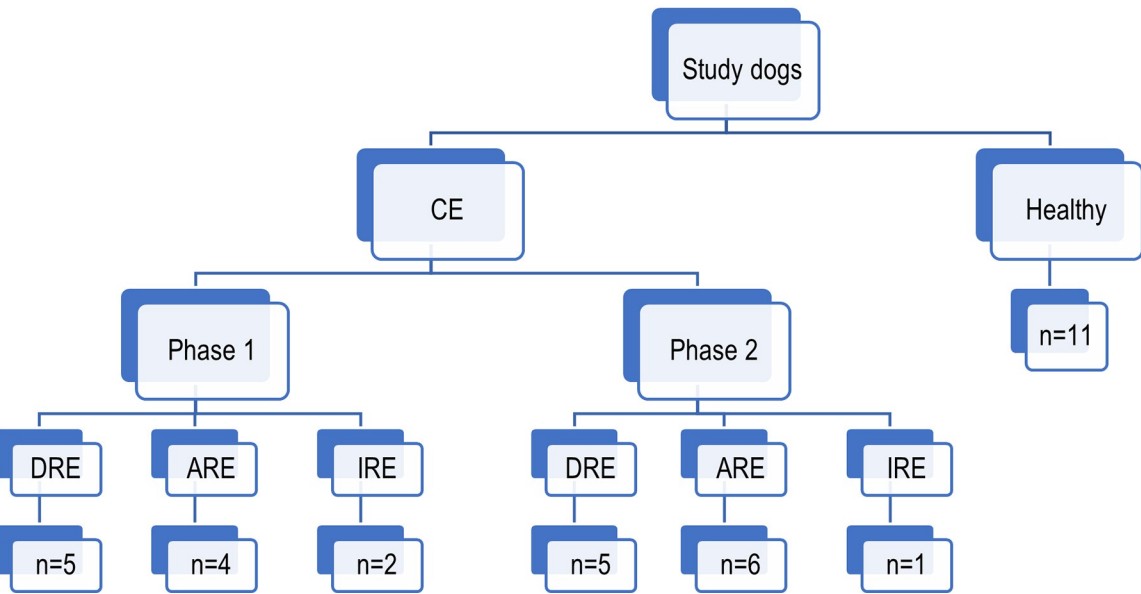

**Fig 1. Breakdown of the study dog population.** CE: Chronic enteropathy. DRE: Diet-responsive enteropathy; ARE: Antibiotic-responsive enteropathy and IRE: Immunosuppressant-responsive enteropathy.

**Table 1. Demographic data of study dogs.**

| | | Healthy | | DRE | | | ARE | | IRE | |
|---|---|---|---|---|---|---|---|---|---|---|
| | | N | Range (Median) | N | | Range (Median) | n | Range (Median) | n | Range (Median) |
| Age | | 11 | 1–11 (5) | 11 | | 1–14 (3.5) | 8 | 2–7 (2.5) | 3 | 5–10 (9) |
| Gender | Female | 7 | | 3 | | | 5 | | 1 | |
| | Male | 4 | | 9 | | | 3 | | 2 | |
| BCS (out of 9) | | 11 | 3–7 (4) | Active | 9 | 2–6 (4) | 5 | 2–5 (4) | 2 | 1–4 |
| | | | | Remission | 7 | 4–5 (5) | 2 | (4–5) | 2 | 4–6 |
| CCECAI | Active | | NA | 10 | | 3–11 (6.5) | 8 | 5–11 (7) | 2 | 5–7 (6) |
| | Remission | | NA | 8 | | 0–2 (0.5) | 7 | 0–2 (1) | 2 | 0 |
| Duration signs (Months) | | | NA | 11 | | 2–24 (12) | 8 | 3–15 (14) | 3 | 8–24 (14) |

DRE: Diet-responsive enteropathy; ARE: Antibiotic-responsive enteropathy and IRE: Immunosuppressant-responsive enteropathy. n: Number of dogs. NA: Not apply. BCS: Body condition score.

## Immunoglobulin coating of faecal bacteria in dogs with chronic enteropathy

In general, active disease was associated with higher estimate proportions of immunoglobulin coating compared to healthy dogs and decreased during clinical remission. Analysis of the bacterial coating according to treatment classification: DRE, ARE or IRE showed that the pre-treatment IgA coating was higher when more treatment modalities were instituted to achieve clinical remission (Fig 2) (S3 Table). IgA and IgG were strongly correlated, meaning that dogs with higher IgA tend to also have higher IgG. A representative staining can be observed in S1 Fig.

## 16S rRNA sequencing immunoglobulin coated population

IgA and IgG-bound bacteria and negative populations isolated during the cell sorting were subjected to 16S ribosomal ribonucleic acid (rRNA) gene sequencing. From a total of 1,400

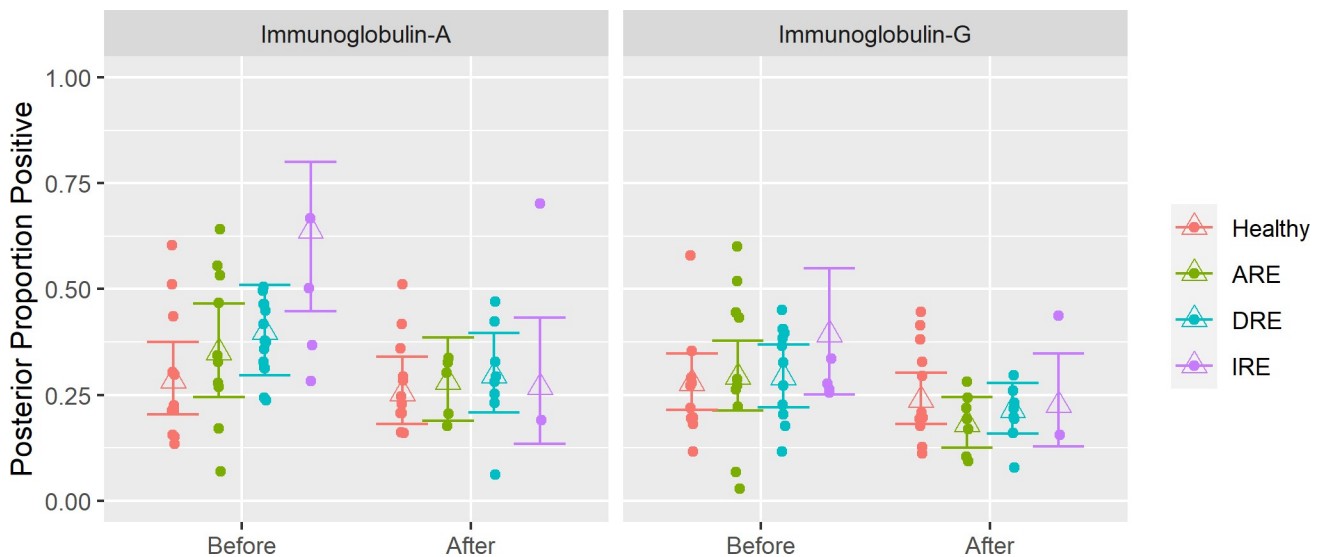

**Fig 2. The posterior predicted relative abundances of immunoglobulin A (IgA) and immunoglobulin (IgG) from healthy dogs and dogs with chronic enteropathy (CE).**

samples (run 1: 728, run 2: 672), including negative and positive controls, input (pre-sort) samples and water collected before each sorting, 1,339 samples (run 1: 719 samples, run 2: 620 samples) were obtained after quality-filtering.

Although numerous precautions were taken to avoid or minimise sample contamination, samples of water collected before the sorting (pre-sorting water) were contaminated with bacterial DNA. At the phylum level, the groups present in the water samples were dominated by Proteobacteria (45%) and Firmicutes (27%), followed by Actinobacteria (9%) and Bacteroidetes (6%). The R package Decontam was used to identify potential contaminants. In healthy dogs as well as in sick dogs with active disease and in remission, the most predominant phylum was Firmicutes and family was *Lachnospiraceae* (Fig 3).

Alpha diversity was analysed using the Shannon index, considering the subject as well as the stage, the Ig class, and the type of CE. Consistent with previous reports [31, 32], we observed that the species diversity of the faecal microbiome was not dramatically altered in

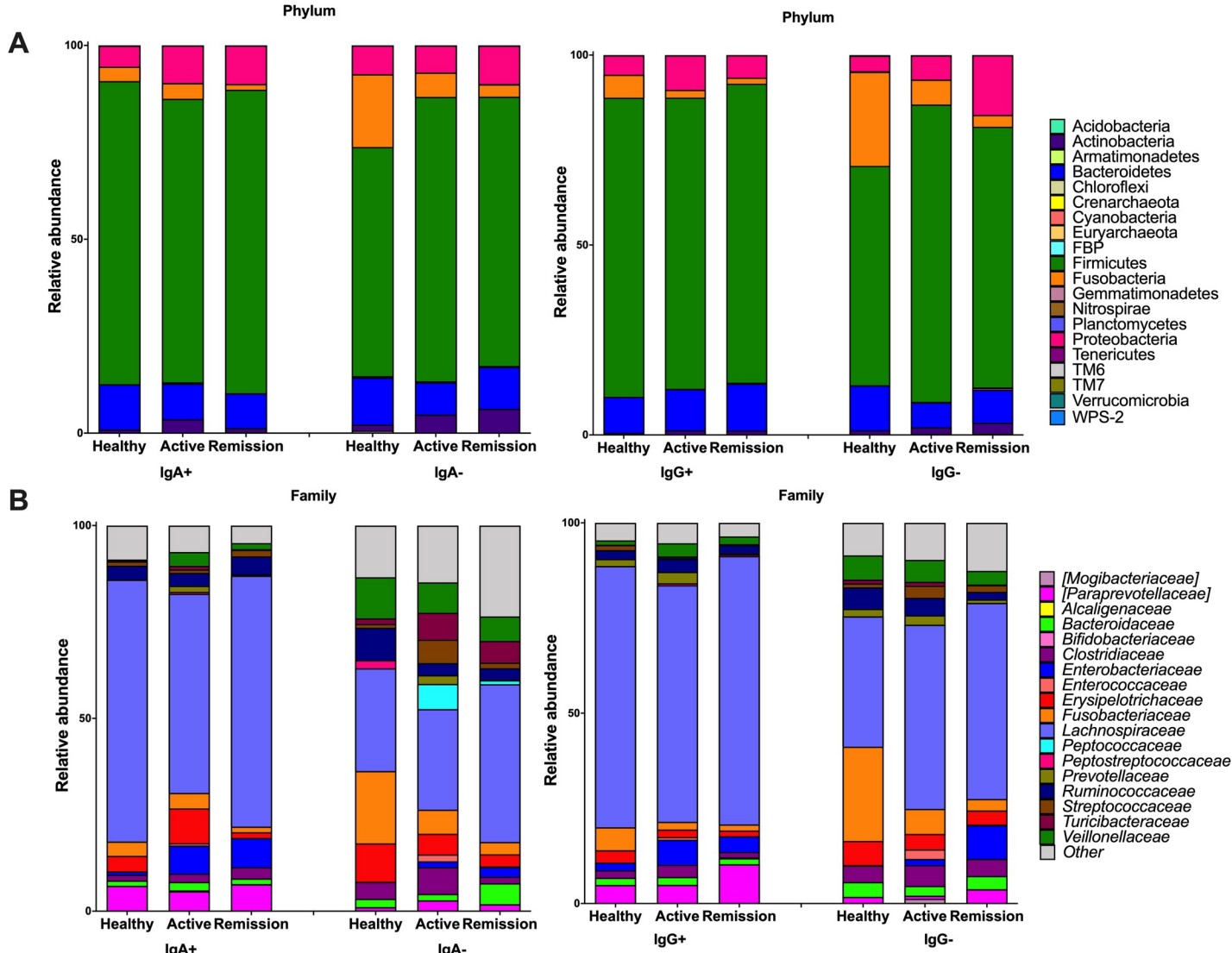

**Fig 3.** Average relative abundance of IgA and IgG positive and negative populations at phylum (A) and family (B) level in healthy dogs and dogs with chronic enteropathy during active disease and remission.

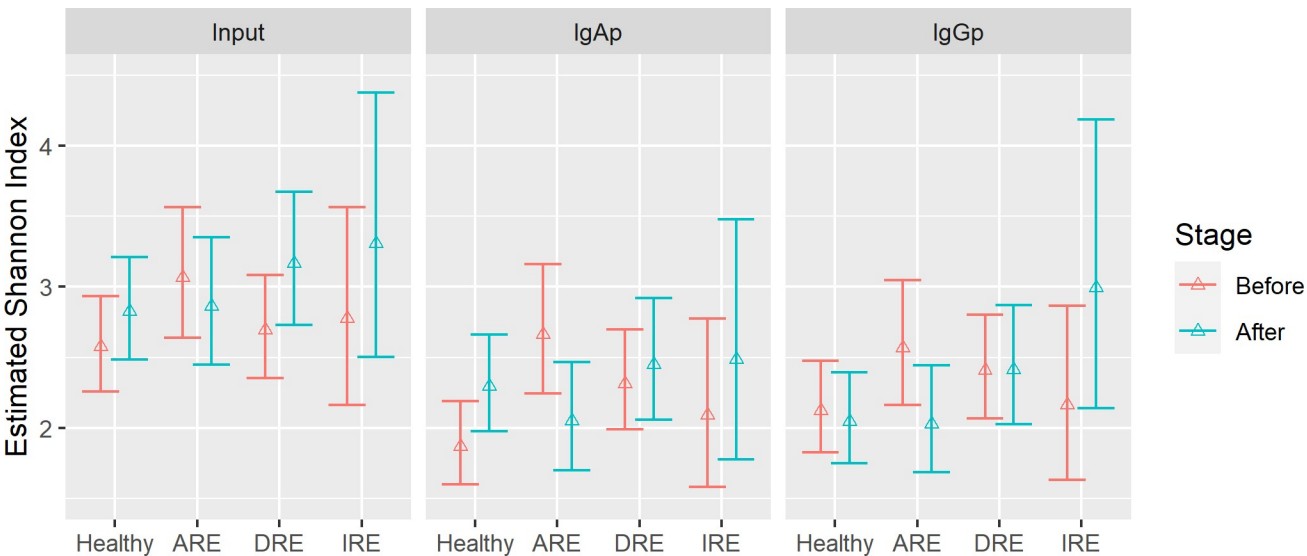

**Fig 4. Estimate of the average of the Shannon index for input (pre-sort) immunoglobulin A (IgA) and immunoglobulin (IgG) from healthy dogs and dogs with chronic enteropathy (CE).** DRE: Diet-responsive enteropathy. ARE: Antibiotic-responsive enteropathy. IRE: Immunosuppressant-responsive enteropathy. 'Before' corresponds to V1 in healthy dogs and active disease in CE dogs. 'After' corresponds to V2 in healthy dogs and remission in CE dogs.

dogs with CE compared to healthy controls, between the different stages of the disease or between different types of CE (marginal $R^2$: 0.24). However, it should be noted that the Shannon index in ARE dogs showed a notable decrease in diversity during the remission period compared to active disease (Fig 4) (S4 Table).

Beta-diversity analysis using Aitchison's distances showed no difference according to health status by Ig coating (PERMANOVA; IgA$^+$ Marginal $R^2$: 0.015, $P$ value: 0.517; IgG$^+$ Marginal $R^2$: 0.016; $P$ value: 0.322, 999 permutations); among the different stages of the disease by immunoglobulin coating (PERMANOVA; IgA$^+$ Marginal $R^2$: 0.018, $P$ value: 0.974; IgG$^+$ Marginal $R^2$: 0.017; $P$ value: 0.998, 999 permutations) or by type of disease. Aitchison's distance was selected for the microbiome β-diversity metric to account for compositionality of relative abundance profiles (S2 Fig) [23].

Analysis of the pattern of coating revealed high inter-individual variability. Hierarchical clustering using Bray-Curtis distances between samples outlined strong structural intraindividual sample proximity and tended to cluster by animal in healthy dogs (S3 Fig), suggesting temporal stability of the Ig coating. Whereas in sick dogs, intraindividual samples were more dispersed. However, in some dogs, all samples clustered; with the majority of these in the DRE group. This applied to IgA$^+$ and IgG$^+$ positive populations (S4 and S5 Figs).

Next, we assessed the microbial differential expression between healthy dogs and dogs with the disease. Enrichment at the family-level taxon in the IgA$^+$/IgG$^+$ fraction was chosen; because at genus and species level, many of the bacteria were not assigned to any group. Assessment of the abundance of the different families in the pre-sort population didn't show an evident clustering pattern between the different states of health or stages of the disease (S6 Fig) and demonstrated the inter-individual variation (Fig 5 and S5 Table).

However, assessment of Ig enrichment ratio across taxa identified a transition from high to low enrichment of *Erysipelitrochaceae* (IgA-G), from low to high enrichment for *Clostridiaceae* (IgA) and from no enrichment to negative enrichment for *Bacteroidaceae* (IgA-G), *Prevotellaceae* (IgA-G) and *Enterobacteriaceae* (IgG) in the DRE group (Fig 6). In the ARE group, we

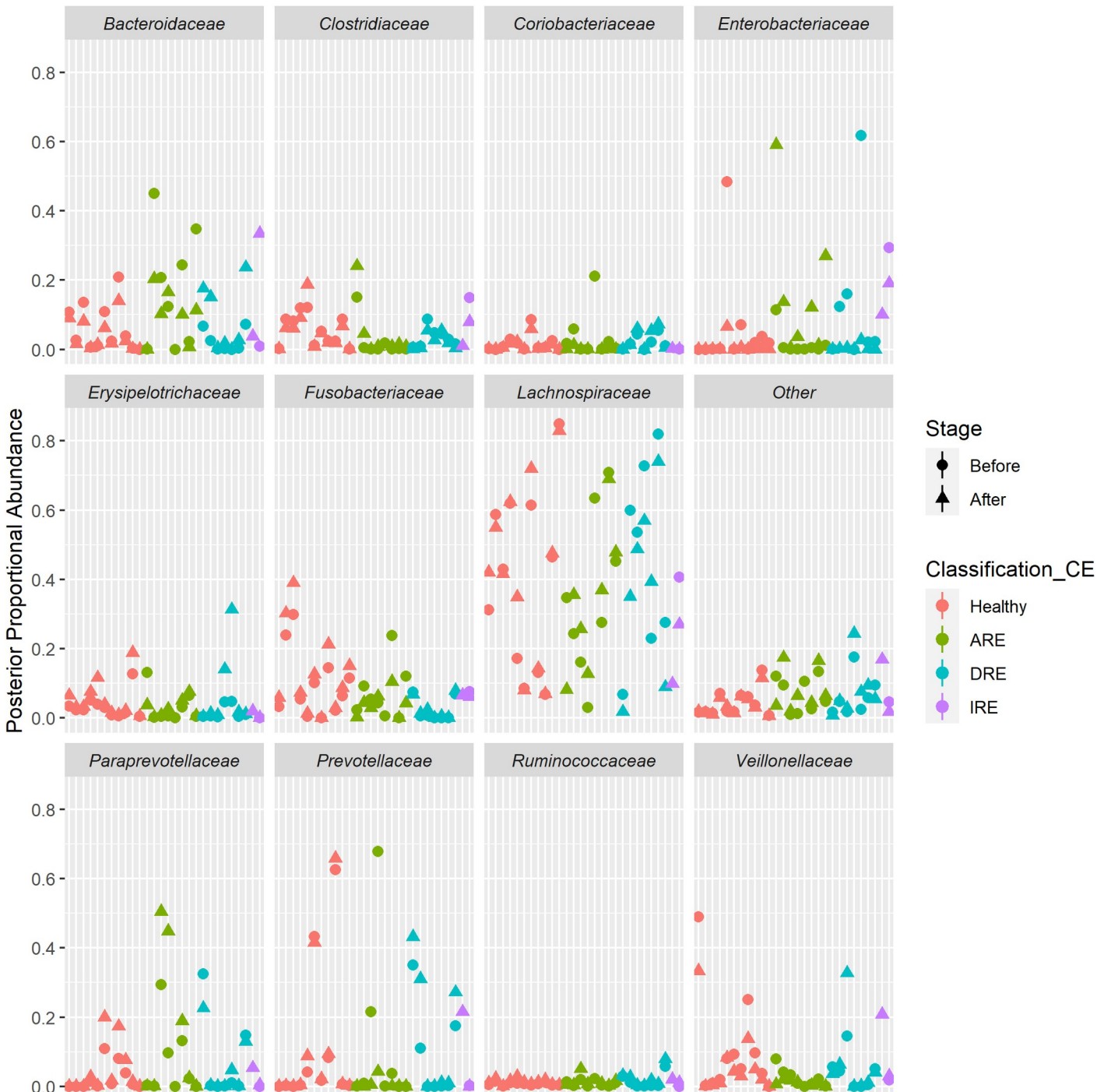

**Fig 5. The subject estimate of the proportional abundance of the different families in the input (pre-sort) population.** DRE: Diet-responsive enteropathy. ARE: Antibiotic-responsive enteropathy. IRE: Immunosuppressant-responsive enteropathy. 'Before' corresponds to V1 in healthy dogs and Active disease in CE dogs. 'After' corresponds to V2 in healthy dogs and remission in CE dogs. Top eleven of the most represented families. Other includes the remaining families.

observed a transition from high to low enrichment of *Erysipelitrochaceae* (IgG), *Enterobacteriaceae* (IgG) and *Clostridaceae* (IgG); and from no enrichment to negative enrichment for *Prevotellaceae* and *Fusobacteriaceae* (IgA) (Fig 6, S6 and S7 Tables).

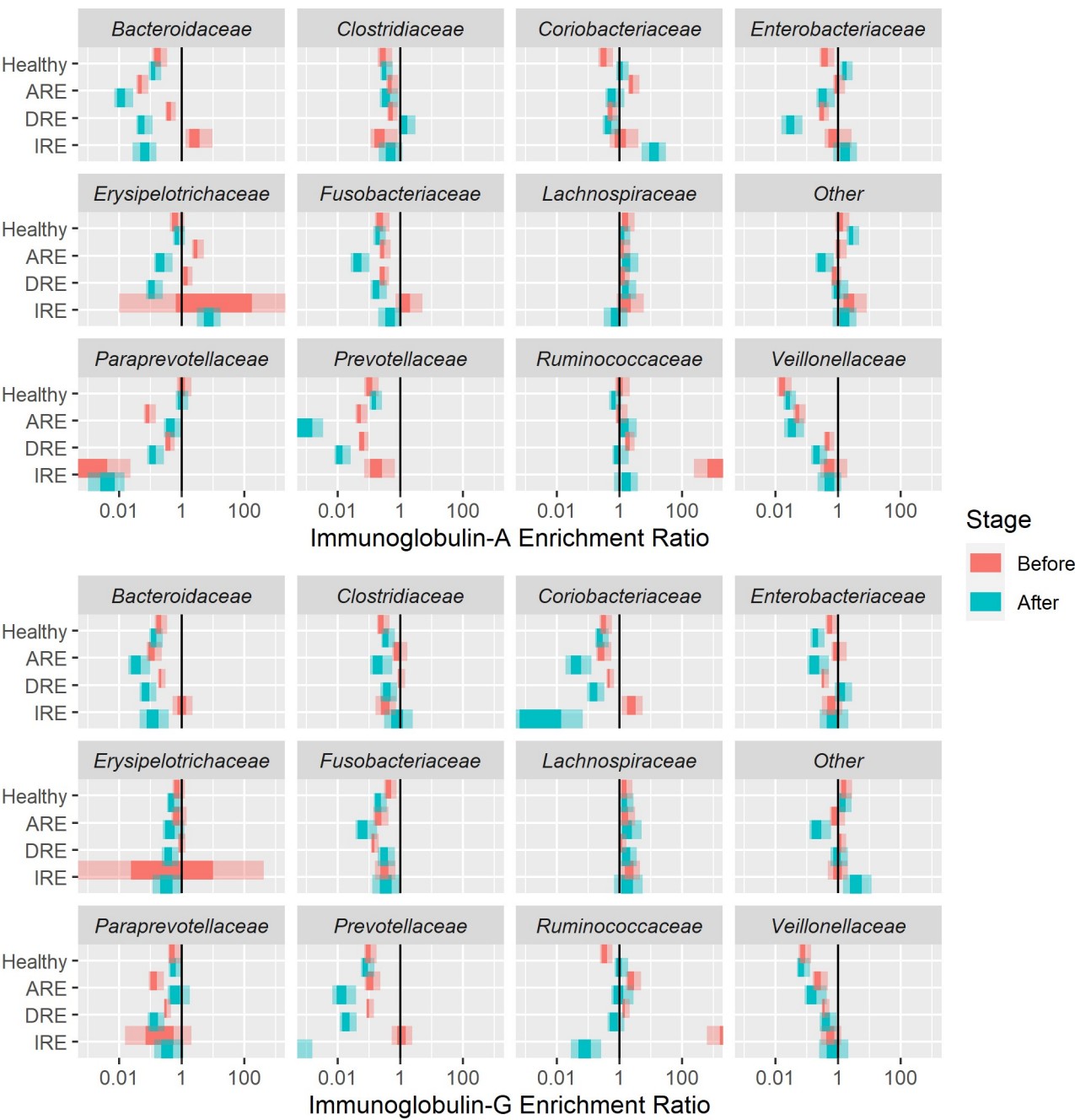

**Fig 6. Immunoglobulin enrichment ratio at family level.** (A) Immunoglobulin A. (B) Immunoglobulin G. DRE: Diet-responsive enteropathy. ARE: Antibiotic-responsive enteropathy. IRE: Immunosuppressant-responsive enteropathy. 'Before' corresponds to V1 in healthy dogs and active disease in CE dogs. 'After' corresponds to V2 in healthy dogs and remission in CE dogs. Top eleven of the most represented families. Other includes the remaining families. The central credible interval corresponds to 50% and the outer interval to 90%.

## Discussion

In this study, we analysed the percentage of faecal bacteria coated with IgA and IgG in healthy dogs and in dogs with CE during active disease (at the time of diagnosis) and shortly after clinical remission (minimum of six weeks without clinical signs) using flow cytometry. As

observed in people, only a minority of bacteria were coated with IgA or IgG in healthy dogs, and the percentages remained stable over the short-term (one-month).

During inflammation, it is expected that concentrations of mucosal Igs will increase to maintain intestinal homeostasis and protect the intestinal mucosa from adherent or invasive bacteria. A decreased intestinal mucosal barrier function may lead to an influx of commensal bacteria and their antigens into the mucosa, with subsequent development of inflammation [33]. However, an increase in Igs could also be an indication of a break in tolerance, with induction or exacerbation of inflammation secondarily [34].

In people with IBD, it has been reported that the population of bacteria coated with Igs is significantly higher in active and remission periods compared to healthy subjects [35]. In line with what has been observed in people [36] and to our initial hypothesis, we observed that dogs with CE had Ig coated faecal bacterial counts higher during active disease compared to healthy control dogs. When we analysed the coating of bacteria based on treatment classification; we observed that the levels of coating (first visit), for IgA in particular, were higher when more treatment modalities were needed to achieve clinical remission. Most of the dogs in our study had DRE and may have had milder manifestations of disease. Unfortunately, with only three dogs classified as having IRE we cannot draw solid conclusions about correlation of Ig coating to severity or being refractory to treatment. Our preliminary observations however could suggest that dogs with more severe disease may have gut bacteria with more immunogenic properties or have a higher dysregulation of the mucosal immune response that requires a combination of different treatments to decrease gastrointestinal inflammation.

In our study population, when CE was in remission, the number of Ig-coated bacteria was lower compared to active disease. During the study period, some dogs were followed at multiple timepoints showing that adequate response to treatment was associated with a decrease in the percentage of coated bacteria and an increase in the percentage of non-coated bacteria. In people with IBD, the amount of IgA and IgG coated bacteria is also positively related to the degree of disease activity and decreases over time [37].

In general, the decrease in the percentage of Ig coating correlated with the improvement of the clinical signs in DRE dogs. However, that was not the case for ARE and IRE dogs, where the Ig coating had the tendency to decrease over time, independently of the clinical response. During antibiotic administration, the gut microbiome may be impacted by an unintended loss of keystone taxa that are critical for maintaining homeostasis and lead to loss of biodiversity [38]. In fact, dogs with ARE exhibited a notable decrease in α-diversity during the remission period compared to active disease, and higher inter-individual distance after treatment.

Glucocorticoids change the local inflammatory microenvironment and are also implicated in the restoration of tight junctions and the intestinal epithelial barrier [39]. These modifications not only influence the intestinal microbiome but also the humoral immune response, altering the pattern of Ig coating. A previous study conducted in healthy dogs receiving prednisolone at a dose of 1.0 mg/kg for 14 days did not observe changes in major bacterial groups of the faecal microbiome (diversity or phylogeny) [40]. However, this scenario may not hold true during intestinal inflammation or at higher prednisolone doses. A recent study in dogs with IBD demonstrated that glucocorticoid therapy was associated with beneficial changes in mucosal microbial community structure and enhanced mucosal epithelial apical junction protein expression [41]. Also, people with IBD receiving a course of corticosteroids showed more microbiome fluctuations than patients with stable medication based on calculations of unweighted UniFrac distance between time points [42].

Thus, it is possible that the decrease of the Ig coating in our dogs with IRE is a result of treatment directly targeting immunogenic bacteria, i.e., those more likely to be coated with Ig, or could reflect an alteration with the humoral response itself. The actual amounts of secreted

Igs were not analysed in our study, but the amount of bacterial coating by IgA and IgG could reflect functional quantity of mucosal Ig.

Although IgA is the main Ig isotype present in the intestine, when invasive bacteria trespass the epithelial border, IgG helps IgA to repel invaders and is a second line of defense [43]. For example, people with Crohn's disease have more IgG-binding gut bacteria than healthy volunteers [44] and it has been also reported that IBD lesions exhibit excessive numbers of IgA$^+$ and IgG$^+$ plasma cells with a remarkable skewing toward IgG production, depending on the severity of inflammation [45]. Like IgA, percentages of IgG coated bacteria were higher during active disease and decreased after treatment and resolution of the clinical signs in our study.

We also analysed 16S rRNA sequences of the sorted populations to test our second and third hypotheses. In healthy dogs, the same phylogenetic groups that have been previously reported in normal faeces were identified [46, 47], and the IgA profile was like the IgG profile.

In contrast to our hypothesis, when we evaluated CE dogs, we did not find a clear difference in α- and β- diversity between stages of the disease or between healthy and sick dogs. Our observations were limited to faeces, and a more severe dysbiosis could be present in the small intestine, however this is a similar finding to other canine studies [31, 32].

Studies in people have detected that serum antibody concentrations exhibit a considerable heterogeneity in microbial specificities among IBD patients; this suggests that rather than a global loss of tolerance against gut bacteria, the response is individual and microbe-specific [48]. We did find a large amount of individual variation in the resident microbiota, so it is likely that the bacterial Ig coating associated with disease differs between dogs [49].

Longitudinal follow-up in microbiome profiles of sick dogs demonstrated how unstable the gut microbiota is during CE. The enrichment profile changed from visit to visit, and hierarchical cluster analysis showed that samples from healthy dogs clustered together or in proximity in each dog, whereas samples in CE dogs were more disperse. This was also in accordance with the proportional abundance predictions in healthy dogs. In people, it has been shown that microbiomes of IBD subjects fluctuated more than healthy individuals and it was characterised by volatile dysbiosis not observed in healthy people [42, 50–53].

Furthermore, in a recent longitudinal study applying multi-omics in people with IBD, the most striking finding was that inter-individual variations explained the majority of variance in all measurement types, suggesting that potentially important changes relevant for disease phenotype and activity are masked within subject-to-subject differences [54]. This may well be the case for our study as well. Thus, rather than trying to identify specific 'causative' or 'associated' bacteria in sick dogs with CE, comparisons within the same dog at different time or disease points appears more appropriate, so as not to over-interpret transient enrichment of a group of bacteria.

Some of the dogs with DRE had microbiota that clustered together between visits. This highlights that DRE may be a milder form of CE where the GI microbiota remains relatively stable like that identified in healthy dogs. Alternatively, the treatment given to DRE dogs may not alter the GI microbiota substantially [32], whereas the treatments given in dogs with ARE and IRE may significantly alter the GI microbiota of dogs. In a longitudinal study in people, diet accounted for a small but significant 3% (false discovery rate (FDR) $P = 7.4 \times 10{-4}$) of taxonomic variation between subjects, and 0.7% (FDR $P = 4.3 \times 10{-4}$) of variation longitudinally [54].

Despite this inter-dog variability and lack of evident changes in bacterial abundance composition in the pre-sort (input) bacteria; when we assessed the ratio of Ig coating, we observed that some families exhibited changes in remission compared to active disease.

We observed that bacteria belonging to the *Erysipelotrichaceae* (DRE and ARE), *Clostridiaceae* (ARE) and *Enterobacterioceae* (DRE and ARE) families were common targets of the Ig response;

and their percentages decreased during the remission period. We also observed negative enrichment of *Bacteroidaceae*, *Fusobacteriaceae* and *Prevotellaceae* (DRE and ARE). Conversely, DRE showed an increase of coating of members of the *Clostridiaceae* family in remission.

Changes in the proportion of *Erysipelotrichaceae* in people with IBD or animal models of IBD have been reported, however the evidence has not been consistent [7, 55–57]. In dogs, one study conducted in German shepherd dogs with chronic intestinal inflammation, demonstrated a significant over-representation of Bacilli and Erysipelotrichi in small intestinal brush samples when compared to healthy Greyhounds [58]. However, other studies have shown a decrease of *Erysipelotrichaceae* in dogs with IBD [59–61].

Increases of members of the *Enterobacterioceae* family have been commonly reported during chronic intestinal inflammation in dogs [60, 62]. Increase in the amount of Proteobacteria may contribute to non-specific mucosal inflammation due to lipopolysaccharides (LPS) as a potent stimulator and possibly predispose the host to a chronic inflammatory disease [63].

In our study, the pattern of enrichment of the *Clostridiaceae* family differed between DRE and ARE dogs. In the DRE group the positive ratio increased during the resolution of clinical signs, whereas in ARE, the negative ratio was the one that increased. A recent study in dogs with DRE observed that remission was associated by a marked decrease in the abundance of pathobionts such as *Escherichia coli* (belonging to Enterobacteriaceae family) and *Clostridium perfringens* (*Clostridiaceae* family) and an increase in the bile acid producer *Clostridium hiranonis* (*Clostridiaceae* family) [31]. Other studies reported decreases of *C. hiranonis* [61, 64] and clostridia [60] during active disease. Conversely, studies have reported that antibiotic use has been associated with increases of *Clostridium*, especially *C. difficile*, a phenomenon attributed to the effect of antibiotics on secondary bile acid concentration within the intestinal lumen [31, 65]. *Clostridiaceae* is a highly diverse family, encompassing genera that are important in nutrient digestibility and immunomodulation; and those that are considered to be pathogenic [66]. Therefore, the differences observed between dogs with ARE and dogs with DRE in our study cannot be determined as analysis was only possible to a family level. Further metagenomic analysis would be necessary to further characterise this difference.

For *Bacteroidaceae*, *Fusobacteriaceae* and *Prevotellaceae* families, most of the studies in dogs have reported lower amounts compared to healthy dogs [61, 64, 67].

There are several limitations in our study. Firstly, there was presence of bacterial DNA in the pre-sorting water, despite our efforts to prevent this. A major challenge to accurately characterise low biomass communities is contamination from external sources. A recent study using flow cytometry has also reported the enrichment of low abundance contaminant reads in their sorted samples [68]. Contamination could potentially occur during sorting and/or library preparation [69]. The bacterial groups identified in the pre-sorting water were excluded from the sample analysis using the R extension Decontam. This may have introduced a negative bias to the study, as some of these bacteria could have been truly present in the samples. The number of bacteria in water and samples could have been quantified by qPCR but was not applied in this study.

Secondly, bacterial DNA content was detected using only SYTO 17. Although useful, other members of the SYTO family (e.g., SYTO 9) are more commonly recommended for bacterial staining. We could not use SYTO 9 as its signal has the same excitation/emission spectrum as FITC, which was the only label option for the antibodies to IgA and IgG.

Thirdly, we focused our study only on the positive population of bacteria. There is a possibility that the negatively coated bacteria could also exert a role in the pathogenesis of intestinal disease. Several breeds of dogs have been identified with IgA deficiency [70], and age and gender effects have not been fully investigated. Thus, future studies should match all control healthy dogs regarding breed, age, and sex with affected dogs.

Fourthly, the lack of a defined phenotypic stratification of the disease in dogs, together with the small number of dogs included in the study and the interindividual variations; hindered the identification of consistent changes in microbial composition.

Fifthly, during statistical analysis we didn't include time as an explicit predictor variable, only included the stage ('before-active' and 'after-remission'); the meaning of 'before' and 'after' differed between groups in a way that we can't mechanistically accommodate; and we had different numbers of measurements from different disease classes owing to the different number of 'before' samples, making the estimates stronger for some subjects than others.

Finally, in people with IBD, the percentages of Ig-coated bacteria return to percentages like those observed in the healthy population only after long-term remission [35]. This phenomenon could also happen in dogs, but we need to analyse long-term remission (i.e., > 1–2 years) samples to determine this. Also, in healthy humans, the response to commensals is restricted to $IgG_2$ responses [71], whereas pathogens induce $IgG_1$ responses. Subsequently, $IgG_1$ antibodies mediate phagocytosis and induce potent proinflammatory pathways, whereas $IgG_2$ is involved in dendritic cell or B-cell activation (tolerance). In dogs, there are four subclasses of IgG ($IgG_1$-$IgG_4$), with $IgG_1$ the most abundant form [72]. One possibility that could be explored in the future is the IgG isotypes that are expressed in dogs with CE. It may be that the isotype IgG response is more important than the overall response.

## Conclusions

Our study showed that immunoglobulin coating of faecal bacteria during CE in dogs was highly individualised despite similar clinical presentation, unstable and was associated with a decrease in the percentage of coating when resolution of the clinical signs occurred. The significance of these changes and the role of these specific groups of bacteria are still unknown. The identified changes could occur secondary to the inflammatory environment of the intestine, or they could directly stimulate the immune system and/or invade the mucosa, contributing to the inflammatory response.

## Supporting information

**S1 Fig. Representative staining of CE during active disease and remission.** For microbial fraction identification the trigger was set up based on side scatter properties. Then, singlets were selected based on SSC-width vs SCC-height. FITC-negative window population was set up in samples containing only SYTO 17 (A). These windows were applied to samples stained with FITC-IgA (B) or FITC-IgG (C) to distinguish the negative population from the positive one.
(TIF)

**S2 Fig. Beta-diversity ordination using the Aitchison distance.** (A) Principal coordinate analysis (PCA) plot IgA healthy versus CE (B) PCA plot IgG healthy versus CE (C) PCA plot IgA stage CE dogs. (D) PCA plot IgG stage CE dogs.
(TIF)

**S3 Fig. Hierarchical clustering of the microbiota profiles in healthy dogs on Bray-Curtis distance.** (A) IgA and (B) IgG populations. Colours represent different dogs.
(TIF)

**S4 Fig. Hierarchical clustering of the microbiota profiles of IgA populations in dogs with chronic enteropathy on Bray-Curtis distance.** Colours represent different dogs.
(TIF)

**S5 Fig. Hierarchical clustering of the microbiota profiles of IgG populations in dogs with chronic enteropathy on Bray-Curtis distance.** Colours represent different dogs.
(TIF)

**S6 Fig. The population estimate of the average proportional abundance of the different families in the input (pre-sort) population.** DRE: Diet-responsive enteropathy. ARE: Antibiotic-responsive enteropathy. IRE: Immunosuppressant-responsive enteropathy. Before corresponds to V1 in healthy dogs and active disease in CE dogs. After corresponds to V2 in healthy dogs and remission in CE dogs. Top eleven of the most representative families. Other includes the rest of the families. The central credible interval corresponds to 50% and the outer interval to 90%.
(TIF)

**S1 Table. Metadata information of dogs with chronic enteropathy.**
(DOCX)

**S2 Table. Metadata information healthy dogs.**
(DOCX)

**S3 Table. Estimate proportions of sorted coated bacteria and credible intervals.**
(DOCX)

**S4 Table. The effect of predictive values on Shannon diversity, with confidence intervals.**
(DOCX)

**S5 Table. The subject estimate values of the proportional abundance of the different families in the input (pre-sort) population and their credible intervals.**
(DOCX)

**S6 Table. Estimates of the immunoglobulin A ratios and their credible intervals.**
(DOCX)

**S7 Table. Estimates of the immunoglobulin G ratios and their credible intervals.**
(DOCX)

## Acknowledgments

The authors acknowledge the support of Stephen Wilcox and Katharina Stracke in the 16S rRNA gene sequencing, Andrew Stent for the histopathology analysis and Leilani Santos for her support during the experiments and all the staff and nurses from the U-Vet hospital at the Faculty of Veterinary and Agricultural Sciences at the University of Melbourne for their help in the collection of samples.

## Author Contributions

**Conceptualization:** Lina María Martínez-López, Caroline Mansfield.

**Data curation:** Lina María Martínez-López, Alexis Perez-Gonzalez, Elizabeth Ann Washington, Andrew P. Woodward, Alexandra Jazmin Roth-Schulze.

**Formal analysis:** Lina María Martínez-López, Alexis Perez-Gonzalez, Elizabeth Ann Washington, Andrew P. Woodward, Alexandra Jazmin Roth-Schulze.

**Funding acquisition:** Lina María Martínez-López, Caroline Mansfield.

**Investigation:** Julien R. S. Dandrieux, Nathalee Prakash, Caroline Mansfield.

**Methodology:** Lina María Martínez-López, Alexis Perez-Gonzalez, Elizabeth Ann Washington, Andrew P. Woodward, Aaron Jex.

**Project administration:** Caroline Mansfield.

**Resources:** Alexandra Jazmin Roth-Schulze, Julien R. S. Dandrieux, Nathalee Prakash, Aaron Jex, Caroline Mansfield.

**Software:** Lina María Martínez-López, Andrew P. Woodward, Alexandra Jazmin Roth-Schulze.

**Supervision:** Thurid Johnstone, Aaron Jex, Caroline Mansfield.

**Validation:** Elizabeth Ann Washington.

**Visualization:** Lina María Martínez-López.

**Writing – original draft:** Lina María Martínez-López, Alexis Perez-Gonzalez, Elizabeth Ann Washington, Andrew P. Woodward, Alexandra Jazmin Roth-Schulze, Julien R. S. Dandrieux, Nathalee Prakash, Aaron Jex, Caroline Mansfield.

**Writing – review & editing:** Lina María Martínez-López, Alexis Perez-Gonzalez, Elizabeth Ann Washington, Andrew P. Woodward, Alexandra Jazmin Roth-Schulze, Julien R. S. Dandrieux, Thurid Johnstone, Nathalee Prakash, Caroline Mansfield.

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
