## [Decision Letter · Decision Letter 0]

8 Jun 2021

PONE-D-21-14847

Hierarchical modelling of immunoglobulin coated bacteria in dogs with chronic enteropathy shows reduction in coating with disease remission but marked inter-individual and treatment-response variability.

PLOS ONE

Dear Dr. Mansfield,

Thank you for submitting your manuscript to PLOS ONE. After careful consideration, we feel that it has merit but does not fully meet PLOS ONE’s publication criteria as it currently stands. Therefore, we invite you to submit a revised version of the manuscript that addresses the points raised during the review process.

This manuscript is worthy of publication after revisions aimed at tightening the text which for now reads a little too much like a PhD dissertation chapter. Reviewer 1 provides suggestions to streamline the manuscript accordingly.

We look forward to receiving your revised manuscript.

Kind regards,

Franck Carbonero, PhD

Academic Editor

PLOS ONE

Additional Editor Comments (if provided):

This manuscript is worthy of publication after revisions aimed at tightening the text which for now reads a little too much like a PhD dissertation chapter. Reviewer 1 provides suggestions to streamline the manuscript accordingly.

Reviewers' comments:

Reviewer's Responses to Questions

**Comments to the Author**

1. Is the manuscript technically sound, and do the data support the conclusions?

Reviewer #1: Partly

2. Has the statistical analysis been performed appropriately and rigorously? 

Reviewer #1: No

3. Have the authors made all data underlying the findings in their manuscript fully available?

Reviewer #1: No

4. Is the manuscript presented in an intelligible fashion and written in standard English?

Reviewer #1: Yes

5. Review Comments to the Author

Reviewer #1: The authors have described a study measuring the immunoglobulin-coated bacteria in dogs suffering from chronic enteropathy. This topic is of interest, but several questions and concerns need to be addressed by the authors. General and specific comments are provided below.

General comments:

1. A key limitation in this study is low animal numbers per group, resulting in very low power. Immune response and microbiota are highly variable among animals. This limitation should be listed first in that area of the Discussion section and an estimate on sufficient numbers in future studies should be estimated.

2. The Methods section is well-described but is very long. If the authors can cite methods to make it shorter, it would be preferred.

3. There are 4 pages of statistical methods in the Methods section, but then there is little to no statistical data/p values given in the Results section, tables, and figures. The authors need to add these key details so that the reader can ascertain what was different (and can be discussed) and what was not.

4. Many section of the Results section should be in the Methods section (lines 406-428; lines 490-494; lines511-513).

5. The Discussion section is much too long for the limited number of differences noted. It should be shortened significantly, focusing on topics/data supported by statistical differences.

6. More technical terms (e.g., observed instead of found or seen, etc.) should be used and grammatical errors (many words capitalized when not appropriate) should be corrected.

Specific comments:

1. Line 37, 132, and elsewhere: “found” and “seen” should be replaced with a more technical term

2. Line 132-134: remained on same diet?

3. Line 170: was pathologist blinded?

4. Line 262: 2.5 not 2,5

5. Line 307: only 1,000 sequences used per sample? Why so low? Why go from 14,000 to 1,000 per sample?

6. Line 307: is there a citation for the Aitchison distance?

7. Line 448-449 and Table 1: why provide both? If sufficient data are provided in Figure 1, Table 2 can be deleted.

8. Line 469-480: shorten and refer to the figure so there is no duplication.

9. Line 485-486: delete sentence. This is anecdotal.

10. Line 494-496: don’t give totals – it is not useful. Give the high-quality sequences used per sample.

11. Line 517 and elsewhere: what statistical value goes along with “decreased”? All differences noted need to be supported by p values. If significant p value does not exist, it is not different and should be stated as such (and not discussed).

12. Line 520: why is it estimated? If Shannon index was calculated, why is it estimated?

13. Line 521 and 524 and elsewhere: why are insignificant words capitalized?

14. Lines 538-541: there is still a lot of clustering within dog in ill animals too. Unless statistics support this statement, it should be revised.

15. Lines 558-565: are there statistics to support these statements?

16. Lines 626-627: do statistics support this?

17. Line 627: do not refer to figures or tables in Discussion. Those should only be in the Results section.

18. Line 634: revise “made in dogs” so it is an appropriate statement.

19. Line 649: trespass is misspelled

20. Lines 662-665: likely due to low power, correct? This needs to be highlighted early on.

21. Lines 683-690: did statistics support this? If not remove or revise.

22. Line 703: which families? Should list specifics here so results can be compared to literature.

23. Lines 733-738, 753-754, and elsewhere: these need to be clearly shown in Results section with appropriate statistics if it is going to be discussed here.

24. Lines 844-847: this is the key issue and should be listed first.

25. Lines 877-883: this is not a conclusion, but future study ideas. Remove and add more information from the current study.

26. Figure 3: estimated? Why not just Shannon index?

27. Supplementary Table 3: predicted – not actual?

6. PLOS authors have the option to publish the peer review history of their article (what does this mean?). If published, this will include your full peer review and any attached files.

Reviewer #1: No

---

## [Author Response · Author response to Decision Letter 0]

30 Jun 2021

Thank you- a response to reviewers has been attached to the submission portal. We think many of the comments are based on statistical queries and so welcome thsi debate.

---

## [Decision Letter · Decision Letter 1]

8 Jul 2021

Hierarchical modelling of immunoglobulin coated bacteria in dogs with chronic enteropathy shows reduction in coating with disease remission but marked inter-individual and treatment-response variability.

PONE-D-21-14847R1

Dear Dr. Mansfield,

We’re pleased to inform you that your manuscript has been judged scientifically suitable for publication and will be formally accepted for publication once it meets all outstanding technical requirements.

Kind regards,

Franck Carbonero, PhD

Academic Editor

PLOS ONE

Additional Editor Comments (optional):

Reviewers' comments:

Reviewer's Responses to Questions

**Comments to the Author**

1. If the authors have adequately addressed your comments raised in a previous round of review and you feel that this manuscript is now acceptable for publication, you may indicate that here to bypass the “Comments to the Author” section, enter your conflict of interest statement in the “Confidential to Editor” section, and submit your "Accept" recommendation.

Reviewer #1: All comments have been addressed

2. Is the manuscript technically sound, and do the data support the conclusions?

Reviewer #1: Yes

3. Has the statistical analysis been performed appropriately and rigorously? 

Reviewer #1: I Don't Know

4. Have the authors made all data underlying the findings in their manuscript fully available?

Reviewer #1: Yes

5. Is the manuscript presented in an intelligible fashion and written in standard English?

Reviewer #1: Yes

6. Review Comments to the Author

Reviewer #1: (No Response)

7. PLOS authors have the option to publish the peer review history of their article (what does this mean?). If published, this will include your full peer review and any attached files.

Reviewer #1: No

---

## [Editor Report · Acceptance letter]

9 Aug 2021

PONE-D-21-14847R1 

Hierarchical modelling of immunoglobulin coated bacteria in dogs with chronic enteropathy shows reduction in coating with disease remission but marked inter-individual and treatment-response variability. 

Dear Dr. Mansfield:

I'm pleased to inform you that your manuscript has been deemed suitable for publication in PLOS ONE. Congratulations! Your manuscript is now with our production department. 

Kind regards, 

on behalf of

Dr. Franck Carbonero 

Academic Editor

PLOS ONE